# Characterization of a Novel African Swine Fever Virus p72 Genotype II from Nigeria

**DOI:** 10.3390/v15040915

**Published:** 2023-04-02

**Authors:** Aruna Ambagala, Kalhari Goonewardene, Lindsey Lamboo, Melissa Goolia, Cassidy Erdelyan, Mathew Fisher, Katherine Handel, Oliver Lung, Sandra Blome, Jacqueline King, Jan Hendrik Forth, Sten Calvelage, Edward Spinard, Douglas P. Gladue, Charles Masembe, Adeyinka J. Adedeji, Toyin Olubade, Nanven A. Maurice, Hussaini G. Ularamu, Pam D. Luka

**Affiliations:** 1National Centre for Foreign Animal Disease, Canadian Food Inspection Agency, Winnipeg, MB R3E 3M4, Canada; 2Department of Medical Microbiology and Infectious Diseases, Max Rady College of Medicine, University of Manitoba, Winnipeg, MB R3E 0J9, Canada; 3Institute of Diagnostic Virology, Friedrich-Loeffler-Institut, Suedufer 10, 17493 Greifswald, Germany; 4Plum Island Animal Disease Center, Agricultural Research Service, United States Department of Agriculture, Greenport, NY 11944, USA; 5College of Natural Resources (CoNAS), Makerere University, Kampala P.O Box 7062, Uganda; 6National Veterinary Research Institute, Vom 930103, Nigeria

**Keywords:** African swine fever, Nigeria, genotype II, West Africa

## Abstract

African swine fever (ASF) is a high-consequence transboundary hemorrhagic fever of swine. It continues to spread across the globe causing socio-economic issues and threatening food security and biodiversity. In 2020, Nigeria reported a major ASF outbreak, killing close to half a million pigs. Based on the partial sequences of the genes B646L (p72) and E183L (p54), the virus responsible for the outbreak was identified as an African swine fever virus (ASFV) p72 genotype II. Here, we report further characterization of ASFV RV502, one of the isolates obtained during the outbreak. The whole genome sequence of this virus revealed a deletion of 6535 bp between the nucleotide positions 11,760–18,295 of the genome, and an apparent reverse complement duplication of the 5′ end of the genome at the 3′ end. Phylogenetically, ASFV RV502 clustered together with ASFV MAL/19/Karonga and ASFV Tanzania/Rukwa/2017/1 suggesting that the virus responsible for the 2020 outbreak in Nigeria has a South-eastern African origin.

## 1. Introduction

African swine fever (ASF) is a deadly contagious hemorrhagic disease of domestic and wild pigs [1]. It is rapidly spreading across the globe with commercial vaccines largely unavailable [2]. The causative agent, ASF virus (ASFV), is a large DNA virus belonging to the family Asfarviridae [3]. The ASFV genome is 170–192 kb long and it encodes 150 to 200 proteins [4,5,6]. Based on the 478 bp fragment corresponding to the C-terminal end of the B646L gene, which encodes the major capsid protein p72, all known ASFV strains are divided into 24 genotypes [4,7,8]. All 24 ASFV genotypes are present in Southeastern Africa [9].

ASF was first reported in Kenya in 1921 [10]. Until the late 1950s, ASF was restricted to the Southeastern parts of sub-Saharan Africa where a sylvatic cycle among warthogs and soft ticks acts as reservoirsof the virus while any introduction into domestic pigs results in a severe and often lethal hemorrhagic fever [11,12]. In 1957, an ASFV genotype I strain from Angola spread to Portugal and subsequently to Spain (1960), France (1964), Italy (1967), Belgium (1985), and the Netherlands (1986) [13]. Later, the virus spread to Malta, Cuba, Brazil, Haiti, and the Dominican Republic [14]. The outbreak was successfully eradicated from the Americas and Europe by 1995, except for the Island of Sardinia which is only now coming to the brink of eradicating the ASFV p72 genotype I strain [15,16]. The second ASF incursion outside Africa was reported in June 2007 in domestic pigs in Georgia [17]. The ASFV strain responsible belonged to ASFV p72 genotype II which was circulating in Mozambique, Madagascar, and Zambia at the time [18]. The virus quickly spread to Armenia (August 2007) and the Russian Federation (November 2007), and entered the wild boar population. Through the wild boar, the virus slowly but steadily spread to neighboring countries in Eastern Europe [Ukraine (July 2012), Poland (February 2013), Belarus (June 2013), Lithuania (January 2014), Latvia (June 2014), Estonia (September 2014), Czech Republic (June 2017), Romania (July 2017), Hungary (April 2018), Bulgaria and Moldova (July 2018)], and most recently into Western Europe [Belgium (September 2018), Greece (February 2020), Germany (September 2020), and Italy (January 2022)] [19,20,21,22]. The virus entered China, the country with the largest pig population in the world, in August 2018 [23,24]. From China ASFV rapidly spread to neighboring countries including Mongolia (January 2019), Vietnam (February 2019), Cambodia (April 2019), Laos (June 2019), Philippines (July 2019), Timor-Leste (September 2019), South and North Korea (October 2019), India (November 2020) and Myanmar (June 2021). ASF recently spread to the Dominican Republic (April 2021) and Haiti (September 2021) [25,26].

ASFV was first introduced to Western Africa through Senegal in 1957, around the same time it was introduced to Portugal [27]. ASFV was isolated from a domestic pig in Senegal in 1959 (Dakar/59) and typed as an ASFV P72 genotype I [27]. Later it spread to Cape Verde, Gambia and Guinea Bissau [28]. The second introduction of ASFV into West Africa was in 1996 in Côte d’Ivoire [29]. By 1997, a number of outbreaks were reported along the West African coast including Benin, Nigeria and Togo, and in 1999 in Ghana and 2003 in Burkina Faso [30,31]. Following these outbreaks, ASF remained endemic in a number of countries in West Africa including Ghana, Nigeria, Senegal, Togo and Guinea Bissau, possibly via free-ranging populations of pigs [31,32]. Recently, a number of ASF outbreaks have been reported in Côte d’Ivoire [33], Ghana (2019, 2020; 2021) [34], Sierra Leone (2020) [35], and Nigeria [36,37]. Despite many ASF outbreaks, until 2020, only the ASFV p72 genotype I strains have been reported from West Africa [38,39].

The first ASF outbreak in Nigeria was reported in mid-1997 in Oko-Oba in the Agege area of Lagos state [40]. The disease was later spread to the Central, Eastern and Southern regions of Nigeria. The virus responsible (98/ASF/NG) was isolated and the complete p72 gene was sequenced (GenBank Accession # AF159503) [41]. The sequence showed 92.2%, 92.4%, and 97.2% homology with previously sequenced ASFV p72 genotype I Ugandan, Dominican Republican and Spanish isolates, respectively. In 2001, an ASF outbreak was reported in Ibadan, Oyo State, affecting over 300 pig farms [42]. Since then ASF remained endemic in the main pig-rearing areas of Nigeria with periodic waves of the disease [43,44]. The virus was also detected in red river hogs (*Potamochoerus porcus*) and warthogs (*Phacochoerus africanus*) in some parts of Nigeria, although, this may be an accidental spillover from domestic pigs [45,46]. ASF has also been detected in asymptomatic domestic pigs indicating possible low virulence mutants circulating in Nigeria. Using genotyping tools, ASFV genotype I strains circulating in Nigeria have been well characterized. Based on the central variable region (CVR) of the B602L gene, 21 variants of ASFV genotype I have been recovered in Nigeria from 1997–2022 [37,47,48].

In February of 2020, increased mortality was observed in the Oke Aro Farm settlement in Lagos state, Nigeria, one of the largest pig farms in Western Africa. At the beginning of the outbreak the fatalities were minimal, but the numbers increased drastically by May 2020, killing over 500,000 pigs and affecting the livelihood of close to 3000 farmers. Four farmers working at the farm died of shock after witnessing the severe mortality caused by the outbreak (https://www.bbc.com/news/world-africa-52993366 - accessed on 10 June 2020). On 6 June 2020, the National Veterinary Research Institute of Nigeria (National Reference Laboratory) confirmed the presence of ASFV-DNA by virus-specific polymerase chain reaction (PCR) in tissues collected from dead pigs at the Oke Aro Farm settlement.

The farm was immediately quarantined and disinfected. Movement control inside the country, official disposal of carcasses, and surveillance outside the containment and/or protection zone were implemented. The outbreak was reported to the OIE (Now WOAH) by the Federal Ministry of Agriculture and Rural Development, Abuja, Nigeria on 16 June 2020. Despite the strict control measures, the outbreak spread to other states quickly. ASF was confirmed in the Ogun, Abia and Edo states in Southern Nigeria and Plateau state in the North-Central region by 23 June 2020 and Delta, a state southeast of Edo by 24 July. By August 2020, ASF was confirmed in 12 out of the 36 states (Figure 1). The outbreak continued and caused losses of over NGN 20 billion naira (USD 43 million), and more than 20,000 jobs associated with the pig industry in Nigeria. Tissue samples collected from four pigs that died of ASF-like disease in Lagos State, the commercial capital of Nigeria, were subjected to genotyping using partial sequences of the genes B646L (p72) and E183L (p54). The study revealed, for the first time, the presence of ASFV P72 genotype II in West Africa [40].

This report describes the further characterization of ASFV isolates from the same outbreak.

## 2. Materials and Methods

### 2.1. Clinical Samples

A total of 17 tissue samples (mixture of spleen, liver, and lung) collected from 3 states (Lagos, Abia and Rivers) between 21 April and 24 July 2020, were submitted to the National Centre for Foreign Animal Disease (NCFAD), Winnipeg, Canada for confirmatory testing and further characterization. In the same shipment, NCFAD also received one tissue sample from a pig showing clinical signs suspected of ASF at a slaughter slab in the Cross River State in Nigeria on 20 June 2019 (Table 1).

### 2.2. PCR and Genotyping

At the NCFAD, 10% (*w*/*v*) tissue homogenates were prepared in sterile PBS from each tissue sample using Precellys 24 homogenizer (Bertin technologies, Rockville, MD, USA). Nucleic acid was extracted using the 5X MagMaxTM Viral/Pathogen Nucleic Acid Isolation kit using the KingFisherTM Flex Purification system (Thermo Fisher Scientific, Waltham, MA, USA). The extracted nucleic acid samples were then tested using ASFV-p72 specific real-time PCR assay [49] with modified cycling conditions to match the fast protocol of the TaqMan™ Fast Virus 1-Step Master Mix. All real-time PCR-positive samples were subjected to genotyping PCR targeting the B646L (p72) and E183L (p54) using qScript XLT One-Step RT-PCR Kit (Quanta Biosciences). For amplification of the 3′ terminal end of the B646L gene, primers p72FP (5′- GGCACAAGTTCGGACATGT-3′) and p72-RP (5′-GTACTGTAACGCAGCACAG-3′) were used [7]. Full-length E183L gene encoding protein p54 was amplified using PPA89 (5′-TGTAATTTCATTGCGCCACAAC-3′ and PPA722 (5′-CGAAGTGCATGTAATAAACGTC-3′) primers [50]. The amplicons were resolved by agarose gel electrophoresis, purified using a QIAquick gel extraction kit (Qiagen, Toronto, ON, Canada) and sequenced using BigDye Terminator chemistry version 3.1 (Life Technologies, Burlington, ON, Canada) on an Applied Biosystems 3130xl Genetic Analyzer (Life technologies, Carlsbad, CA, USA).

### 2.3. Virus Isolation and Titration

Virus isolation was attempted on samples with Ct value less than 30, following the NCFAD standard operating protocol for African swine fever virus isolation [51]. Briefly, primary pig leukocyte (PPL) cultures were prepared using heparinized whole blood collected from a healthy pig. Leukocytes were separated using 6% *w*/*v* dextran (Sigma-Aldrich Canada Co., Oakville, ON, USA) solution and re-suspended in RPMI (Thermo-Fisher Scientific, Waltham, MA, USA) supplemented with 5% fetal bovine serum (Thermo-Fisher Scientific), 1× Glutamax (Thermo-Fisher Scientific) and 5 mg/mL gentamicin (Gibco, Grand Island, NY) at 10^6^ cells /mL. Washed red blood cells from the same animal were added to the leukocyte cultures to give final RBC concentration of 0.4% *v*/*v*. The leukocyte +RBC suspension was then plated in 24-well plates at 1 mL per well. After 72 h at 37 °C in a 5% CO_2_ incubator, 200 µL of 10-fold dilutions of 10% tissue suspensions were inoculated into duplicate wells. The plates were then incubated for 5–7 days at 37 °C in a 5% CO_2_ incubator, and were examined daily for hemadsorption and cytopathic effects. On the 7th day, the plates were frozen at −20 °C for at least 4 h and thawed at 4 °C. The supernatant from each well was then subjected to ASFV real-time PCR, as described above. The supernatant from real-time PCR-positive samples was subjected to virus titration on primary porcine alveolar macrophage (PAM) cultures. Briefly, 10-fold dilutions of the isolated viruses were inoculated into 90% confluent primary porcine alveolar macrophage (PAM) cells in MEM supplemented with 1% Gentamicin, 1% Glutamax and 2% FBS. Following 3 days of incubation at 37 ℃ and 5% CO_2_, plates were fixed and stained with an anti-ASFV polyclonal pig serum, and HRP-conjugated anti-pig mouse monoclonal antibody (Sigma).

### 2.4. Whole Genome Sequencing, Assembly and Annotation of the ASFV Genome

The ASFV Nigeria/RV502/2020 isolate (hereafter called ASFV RV502) was selected for whole genome sequencing. Total nucleic acid was extracted from PPL cultures (2nd passage) using the 5× MagMax^TM^ Viral/Pathogen Nucleic Acid Isolation kit (ThermoFisher). For Illumina sequencing, ASFV genome in the sample was enriched using a custom myBaits^®^ ASF target capture kit (Daicel Arbor Biosciences, Ann Arbor, MI, USA) and the sequence libraries were prepared using Nextera XT DNA Library Preparation Kit (Illumina, San Diego, CA, USA). The custom myBaits^®^ ASF target capture kit was designed based on 1614 ASFV sequences (19 sequences ranged from 170,101 bp–193,886 bp, and the remaining 1595 sequences ranged from 84 bp–55,098 bp). Sequencing was performed on an Illumina MiSeq instrument at the NCFAD using a V2 flow cell with a 500-cycle reagent cartridge (Illumina). The same nucleic acid extract was subjected to Nanopore sequencing using the GridION X5 (Oxford Nanopore Technologies, MA, USA). For library preparation, the Rapid PCR barcoding kit SQK-RPB004 (Oxford Nanopore Technologies) was used and the samples were run on R9.4 flow cells (Oxford Nanopore Technologies). The raw sequencing signals were processed and converted into the FASTQ format using Guppy base caller version 4.0.11 [52].

To obtain additional sequence data, nucleic acid extracted from 10% (*w*/*v*) tissue homogenate in PBS and whole blood from 1 of the 2 pigs that died after inoculation with ASFV RV502 (described below) was also subjected to Illumina and Nanopore sequencing as described above. In the end, all the data were pooled for analysis.

### 2.5. Phylogenetic Analysis

A total of 162 whole genome sequences (123 curated sequences from Bao et al., 2022 [5], and 30 whole genome sequences recently (between 2021-11-09 to 2022-12-07) uploaded into GenBank (https://www.ncbi.nlm.nih.gov/genbank - accessed on 7 December 2022) as well as 8 additional sequences from the Center of Excellence for African Swine Fever Genomics (https://asfvgenomics.com/GenomeReference.html - accessed on 7 December 2022)) were used in the phylogenetic analysis (Appendix A). A multiple alignment of the sequences was carried out using MAFFT 7.505 (https://mafft.cbrc.jp/alignment/software/) with LAST (http://last.cbrc.jp). The evolutionary history was inferred using the maximum likelihood method with 1000 bootstrap replications and the resulting tree was visualized using ggtree [53]. Metadata including the p72 genotype, host, origin, and year collected was obtained from GenBank records or the literature.

For sero-grouping, full-length EP402R sequences from 156 out of the 162 whole genome sequences were used. A total of 6 out of 162 whole genome sequences did not have EP402R gene sequences. Nine additional previously typed sequences were obtained from Sereda et al., 2022 [53].

## 3. Results

### 3.1. PCR and Genotyping

The total nucleic acid extracted from all tissue samples received from Nigeria tested positive by the ASFV-specific real-time PCR assay at the NCFAD (Table 1). The Ct values varied from 18 (sample #17 from Rivers state) to 39 (sample #14 from Abia state). All samples were then subjected to p72 genotyping PCR, and all except #3, #10, #13 and #14, produced a PCR amplicon around 478 bp. The same samples yielded 676 bp fragments (full-length p54) with the p54-specific primers. Sanger sequence data confirmed that all ASFV strains except CR060T belonged to the ASFV p72 genotype II and p54 genotype II. The CR060T strain belonged to the p72 genotype I and the p54 genotype 1a.

### 3.2. Virus Isolation

All samples with Ct values less than 30 were inoculated into porcine primary leukocyte cultures for virus isolation using the standard protocol at the NCFAD. The HAD-positive virus was isolated from samples #6 (LA10), #8 (LA30), and #9 (LA31) from Oke-Aro in Lagos state, #15 (AB26) from Umahia in Abi state, and #17 (ASFV RV502) from Obio in Rivers state. Virus isolation from other samples including #18 (CR060T) was not successful. No non-HAD viruses were detected.

### 3.3. Whole Genome Sequencing, Assembly and Annotation of the ASFV Genome

The Illumina Miseq instrument generated 2,757,042 total reads, and Nanopore GridION X5 generated 2,046,988 reads over 2 separate runs. Trimmed Illumina MiSeq and Nanopore GridION X5 reads were used for de novo assembly of the genome using CLC Genomics Workbench (Qiagen). The contig was mapped to the Georgia-2007/1 genome (GenBank # NC_044959). The complete genome of ASFV RV502 was annotated using CLC Genomics Workbench (Qiagen) and submitted to GenBank (Accession number: OP672342). The complete genome assembly of ASFV RV502 generated a genome of 185,319 bp with a GC content of 38.5. When the sequence was used for BLASTN search at GenBank, it showed 99.92% and 99.1% nucleotide identity (95% coverage) with ASFV Tanzania/Rukwa/2017/1 (GenBank: # LR813622.1) and ASFV MAL/19/Karonga (GenBank# MW856068.1) isolates, respectively. ASFV RV502 whole genome sequence showed only 99.86% nucleotide identity (99% coverage) to the ASFV Georgia 2007/1 and 99.85% identity and 100% coverage with ASFV Timor-Lester/2019/1 (Timo-Lester), Belgium/Etalle/wb/2018, ASFV Wuhan 2019-1 (China), IND/AR/SD -61/2020 (India).

The whole genome of ASFV RV502 when compared to Georgia 2007/1, has a deletion of 6535 bp between the nucleotide positions 11,760 and 18,295 (positions related to ASFV RV502) of the genome (Figure 2).

The observed deletion occurs between the MGF 110-8L and MGF 360-6L genes, resulting in the deletion of 14 genes (MGF 110-8L, MGF 110-XR, ACD 00190, MGF 110-9L, ACD 00210, MGF 110-10L-14 L, G ACD 00240, MGF 110-12L, MGF 110-13La, MGF 110-13Lb, ACD 00270, MGF 360-4L, ACD 00300 and G ACD 00350). Additionally, compared to Georgia 2007/1, there is a 688 bp deletion that resulted in an in-frame fusion between MGF 110-3L and MGF 110-4L. Interestingly, there is a single nucleotide insertion that occurs within the MGF 360-1L gene, and this insertion changes MGF360-1L to the KP360L gene. This is the first report where KP360L is found in a genotype II strain. At the 3′ end of the genome there is an apparent reverse complement duplication of the 5′ end of the genome that begins in the middle of MGF 360-21R, creating an in-frame fusion between MGF 360-21R and 2L, and continues to include a duplication of KP360L (labelled as KP360R). None of these deletions, fusions or duplications are present in the published genome sequences of ASFV Tanzania/Rukwa/2017/1 and ASFV MAL/19/Karonga strains.

### 3.4. Phylogenetic Analysis

The ASFV RV502 whole genome (OP672342) grouped with p72 genotype II viruses and were clustered together with ASFV MAL/19/Karonga (MW856068) and ASFV Tanzania/Rukwa/2017/1 (LR813622) suggesting that ASFV RV502 likely derived from a virus responsible for the outbreaks in Malawi and Tanzania (Figure 3). A separate evolutionary cluster was also observed with LS478113 (ASFV Estonia 2014), MH910496 (ASFV strain Georgia 2008/2) and two isolates (MT872723 and MT882025) from Vietnam. The large deletion of 14,560 base pairs at the 5′ end, and genome reorganization by duplication in ASFV Estonia 2014 diverged it from the rest of the ASFV genotype II sequences.

Phylogenetic analysis of the full-length EP402R gene confirmed that ASFV RV0502 belongs to serogroup 8, and it clustered with ASFV MAL/19/Karonga and ASFV Tanzania/Rukwa/2017/1 confirming the results of the whole genome analysis (Appendix A).

## 4. Discussion

Nigeria has an estimated population of over seven million pigs which is differentiated into mostly free-roaming/extensive, semi-intensive and intensive production systems in pig-producing areas in Northern Nigeria and intensive commercial/communal pig estates in Southern parts of the country [54]. Several factors have been identified as drivers of ASF in Nigeria including unregulated live pig movement, poor biosecurity measures at farms and trading at live pig markets [43]. However, despite several studies there is limited information on the epidemiological drivers of ASF in Nigeria. Likewise, the genetic information of the circulating ASFVs in the Nigerian pig population is also limited. Additionally, no whole genome sequences of ASFVs responsible for outbreaks in Nigeria are available in the GenBank. From West Africa, only the ASFV Benin 97/1 pathogenic isolate is the only whole genome sequence deposited in the GenBank [55].

The 2020 ASF outbreak killed close to half a million pigs in Nigeria. Based on partial ASFV genome sequences, obtained from 4 tissues collected from a pig farm in Oko Oba, Lagos, an ASFV genotype II isolate was confirmed in this outbreak [39]. In this study we evaluated 17 additional tissue samples collected between 21 April, 2020 and 1 May 2020 from pig farms in 3 states, Lagos, Abia and River. In addition, we also evaluated a sample (CR060T) collected from a sick pig at a slaughter slab from Ikom (Cross River State) on 20 June, 2019. ASFV in all the samples except CR060T, belonged to the ASFV p72 genotype II. The isolate from the ASFV RV502 sample, named ASFV Nigeria/RV502/2020, was subjected to whole genome sequencing. The sequence showed 99.92% and 99.1% nucleotide identity with ASFV isolates from the recent outbreaks from Tanzania and Malawi, suggesting the possible introduction of ASFV genotype II from Southeastern states to Nigeria in West Africa. The ASFV RV502 showed only 99.86% nucleotide identity to the ASFV Georgia 2007/1 and 99.85% identity to ASFV Timor-Lester/2019/1 (Timo-Lester), Belgium/Etalle/wb/2018, ASFV Wuhan 2019-1 (China), and IND/AR/SD -61/2020 (India) whole genome sequences.

Due to increased investments and work opportunities for migrants in the region, there has been an increased migration of foreign nationals from Southeastt Asia and Europe into Nigeria, particularly Lagos State, the commercial capital of Nigeria. Therefore, when the 2020 ASF outbreak happened, a possible introduction of highly virulent ASFV genotype II isolates from Asia or Western Europe was suspected [39]. However, the phylogenetic analysis of whole genome sequences clearly shows that the ASFV RV502 responsible for the 2020 outbreak in Nigeria is more similar to ASFV isolates from Malawi and Tanzania, than those from Asia and Western Europe.

Until 2020, ASFV genotype II has been reported only in South and East African countries. These countries include Madagascar (1998), Mozambique (2006), Mauritius (2007), Tanzania (2010, 2011 and 2017), Malawi (2011 and 2019), Zimbabwe (2015) and Zambia (2013 and 2017) [32,56,57]. Despite a large number of ASFV genotype II outbreaks in South and East Africa, only two whole genomes, Tanzania/Rukwa/2017/1 and ASFV MAL/19/Karonga, have been reported. The 2 whole genome sequences show 99.97% nucleotide identity to each other. Considering the sequence similarity and close geographical location of the two outbreaks, the transboundary spread of ASFV between Tanzania and Malawi was suggested [57].

Epidemiological data collected from farmers during the 2020 outbreak investigations revealed that the outbreak may possibly have started in 2019, but due to non-reporting by farmers and weak veterinary and surveillance systems in Nigeria, the outbreak was not detected early. Furthermore, the disease was initially restricted to Lagos State in early 2020, but started spreading to the Eastern and Northern parts of Nigeria following the easing of COVID-19 lockdown restrictions via the movement of live ASFV-infected pigs mostly from Oke-Oro in Lagos State. Genetic typing tools are very important for tracing and tracking the source and route of the spread of infection. This study suggests, ASFV genotype II may have an ancestral source in East or Southern Africa, but the exact route of introduction into Nigeria is not clear. The farmers from the Oke-Oro pig farming estate, import pedigree pigs from other countries such as South Africa, Israel, China and Belgium, and illegal movements of live pigs and pig products between countries continue to be a problem in Africa.

The ASFV RV502 has a 6. 3 Kb deletion in the 5′ end and of its genome and the deleted area overlaps with the large 15 Kb deletion observed in the partially attenuated ASFV Estonia 2014 (p72 genotype II). We inoculated ASFV RV502 into 2 weaned piglets oro-nasally (2 × 10^5^ TCID50 in 2 mL, 1 ml orally and 0.5 mL in each nostril), and both animals developed a fever and viremia within 3 dpi. Moreover 1 of the 2 piglets was euthanized on 6 days post-infection (dpi) as it reached the humane end-point. The second pig was also euthanized on the same day, as a single pig was not allowed in animal pens at the NCFAD according to the Institutional animal care guidelines. The pig reached humane endpoint exhibited gross lesions similar to those described for highly virulent ASFV strains [58,59,60]. The second pig showed only mild hemorrhagic lymphadenopathy, likely because it was euthanized prematurely. Based on the highly virulent phenotypic outcome observed during the 2020 ASFV outbreak in Nigeria and the results from the animal experiment conducted at the NCFAD, it is highly unlikely that ASFV RV502 is attenuated, but a larger experimental study is required to confirm the virulence of ASFV 502.

## 5. Conclusions

In conclusion, this study documents the report of the whole genome of ASFV II in West Africa and the second whole genome of circulating ASF viruses in the West African sub-region. The phylogenetic analysis suggests that ASFV RV502 has an ancestral source in East and Southern Africa. This study confirms the co-circulation of ASFV genotypes I and II in Nigeria, which complicates the ASF control measures. Therefore, a successful ASF vaccine in Nigeria, needs to protect pigs against both genotype I and II strains. Nigeria is one of the top pig-producing countries in Africa. Over the last 10 years, pig production in Nigeria has increased by 40%, and in 2019, Nigeria produced over 298,000 metric tons of pork according to the Pig Farmers Association of Nigeria (PFAN). The appearance of an ASFV genotype II virus, most likely from Southeastern Africa highlights the possible illegal movement of live pigs and pig products between countries in Africa and complicates the efforts to prevent and control ASF spread in the region. To control ASF in Nigeria, it is therefore recommended that veterinary and surveillance systems should be strengthened, and the importation and movement of pigs should be strictly regulated.

## Figures and Tables

**Figure 1 viruses-15-00915-f001:**
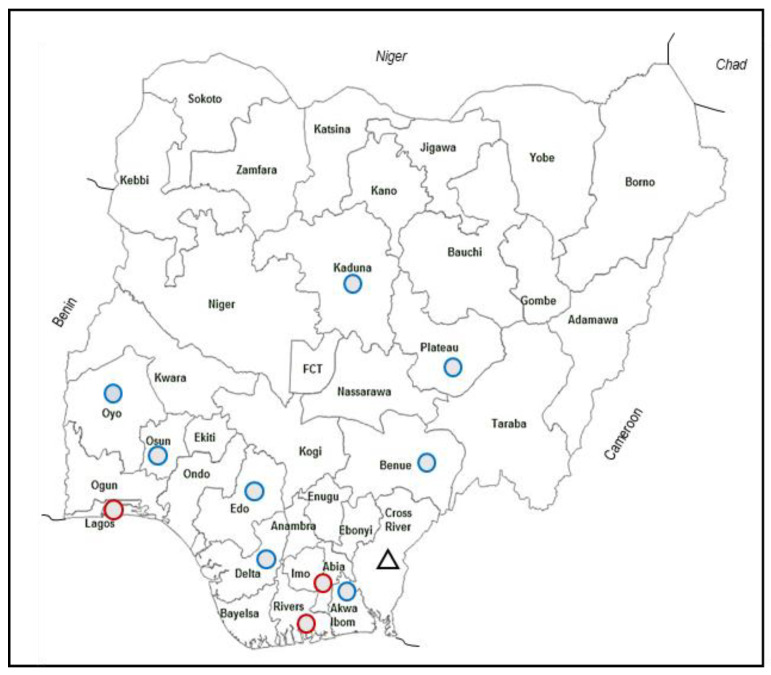
ASF outbreak map. Affected states based on 2020 WAHIS WOAH reports are marked with blue circles. The states in which the samples were tested at the NCFAD are marked with red circles. The Cross River state that reported ASFV genotype I strain is marked with a triangle.

**Figure 2 viruses-15-00915-f002:**
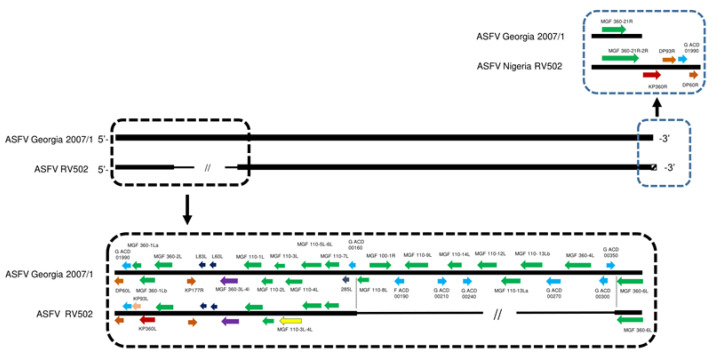
Diagram showing the sites of deletion and reorganization in the ASFV RV502 genome compared to the ASFV Georgia 2007/1. ASFV RV502 genome is lacking 6535 bp between the nucleotide positions 11,760–18,295 of the genome which includes 14 genes. Additionally, a 688 bp deletion resulted in an in-frame fusion of MGF 110-3L and 110-4L (yellow arrow). A single nucleotide insertion within the MGF 360 1L gene, changed MGF360 1L to the KP360L (red arrow). At the 3′ end of the genome, there is an apparent reverse complement duplication of the 5′ end of the genome, creating an in-frame fusion between MGF 360-21R and 2L, and continues to include a duplication of KP360L (labelled as KP360R).

**Figure 3 viruses-15-00915-f003:**
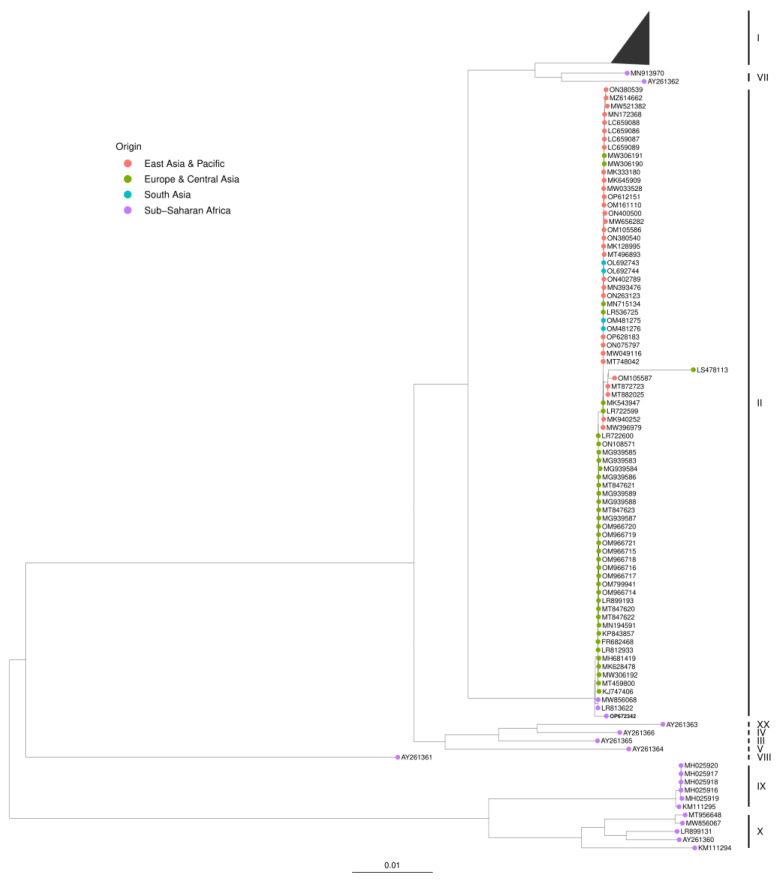
Maximum-likelihood (ML) phylogenetic tree of 162 ASFV whole genome sequences. Sequences were aligned in MAFFT 7.505 with 1000 bootstrap followed by phylogenetic tree construction using the IQ-TREE. The phylogenetic tree was then visualized using ggtree. The 2020 and 2019 ASFV Nigerian strains sequenced are represented in blue. For clarity, only a limited number of ASFV p72 genotypes are depicted. All sequences belonged to Genotype I are collapsed, the sequences representing different countries are color coded, and the ASFV RV502 sequence (OP672342) is in bold.

**Table 1 viruses-15-00915-t001:** History and the summary of laboratory results for the samples tested at the NCFAD. * Sample subjected to whole genome sequencing. (# = Number).

Sample	NVRI Lab. #	Sampling Location	Collection Date	History & Clinical signs	RT-PCR (Ct)	p72 Genotype	Virus Isolation
1	LA4	Oke-Aro, Lagos	2020-05-01	Herd size: 120, 119 died, fever, anorexia, reddened skin	22.74	II	No
2	LA5	Oke-Aro, Lagos	2020-05-01	Same herd as above	22.21	II	No
3	LA7	Oke-Aro, Lagos	2020-05-01	Herd size 134, all died, fever, anorexia, reddened skin	30.48	-	No
4	LA8	Oke-Aro, Lagos	2020-05-01	Herd size 134, all died	34.45	-	No
5	LA9 *	Oke-Aro, Lagos	2020-05-01	Herd size: 163, 111 died	19.94	II	No
6	L10	Oke-Aro, Lagos	2020-05-01	Herd size: 163, 111 died	26.1	II	Yes
7	LA11	Oke-Aro, Lagos	2020-05-01	Same heard as above	29.35	II	No
8	LA30	Oke-Aro, Lagos	2020-04-29	Herd size: 124, 71 died	23.25	II	Yes
9	LA31	Oke-Aro, Lagos	2020-05-01	Herd size: 105, 20 died	25.16	II	Yes
10	LA32	Oke-Aro, Lagos	2020-05-01	Herd size: 90, 54 died	30.58	-	No
11	LA33	Oke-Aro, Lagos	2020-05-01	Herd size: 80, 15 died	24.31	II	No
12	LA34	Oke-Aro, Lagos	2020-05-01	Herd size: 150, 70 died	19.25	II	No
13	LA35	Oke-Aro, Lagos	2020-05-01	Herd size: 129, 20 died	33.23	-	No
14	AB10	Umahia, Abia	2020-04-21	Herd size: 50, all died. fever, cyanosis and depression	39.23	-	No
15	AB26	Umahia, Abia	2020-04-21	Herd size: 87, all died, fever, cyanosis, depression	19.85	II	Yes
16	AB 40	Umahia, Abia	2020-04-21	Herd size: 140, All died or culled, cyanosis, depression	20.99	II	No
17	RV502 *	Obio, Rivers	2020-07-24	Herd size:96, 60 dead, dullness, anorexia	18.74	II	Yes
18	CR060T	Ikom, Cross River	2019-06-20	Samples collected at a slaughter slab	22.51	I	No

## Data Availability

All data related to this study will be made available upon request.

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
