# Peer review of "Characterization of a Novel African Swine Fever Virus p72 Genotype II from Nigeria"

_viruses, 2023, doi:10.3390/v15040915_

Round 1
Reviewer 1 Report
Article entitled “Characterization of a novel African swine fever virus p72 geno-2 type II from Nigeria” has scientific value.
Comments
A quality article providing important and interesting information about the evolution of the ASPV virus.
Performed on a good scientific basis with adequate methods, the results obtained provide important scientific information.
There are only a few minor remarks
1. The description of the pathological picture at autopsy of pigs infected with the ASFV genotype 2 has previously been described in the scientific literature, but the authors do not have references to earlier works. There is also no explanation for the need for the study. As far as one can judge, there are no differences in the pathology in pigs infected with the ASF virus genotype 2 described by the authors and previously published.
2. An experiment conducted on two pigs (1 for each infection variant) cannot be considered reliable due to the insufficient amount of experimental data.
3. It should also be noted that autopsy data are practically not discussed in the Discussion section.
4. Present only gross autopsy, no histology, probably due to analogy with previously described cases it was not necessary?
5. In my opinion, it is better to remove the part of the article on autopsy, replacing it with the statement: the pathology did not differ from the previously described forms of ASF caused by the genotype 2.
6. No reference to virus isolation protocol on porcine primary leukocyte culture
Reviewer 2 Report
The manuscript submitted by Ambagala et al. entitled "Characterization of a novel African swine fever virus p72 genotype II from Nigeria" aims to evaluate the genome of a ASFV II in West Africa and second whole genome of circulating ASF viruses in the West African sub region. the authors showed that ASFV Nigeria is a virulent virus with an ancestral source in east and southern Africa and confirmed the co-circulation of ASFV genotype I and II in Nigeria, a top pig producing countries in Africa. Thus, the results reported are very relevant and I anticipate a great impact of them in the field.
However, the following aspects should be improved before the publication of the work:
Line 42 - the missing of effective ASFV vaccines should be supported by the a recent work (https://doi.org/10.1080/22221751.2022.2108342)
Lines 58-63 and 64-67 - please add the dates of ASF introduction in the different countries
Line 69 - reference 24 should be replaced by https://doi.org/10.1016/j.virol.2014.10.034, which details the ASF introduction in Portugal (or at least added).
Line 113 - convert the value in USD and/or EUR.
In conclusion section, please detail the putative consequences of the genome mutations detected in a vaccination perspective view
